# Performance Comparison of Five Methods for *Tetrahymena* Number Counting on the ImageJ Platform: Assessing the Built-in Tool and Machine-Learning-Based Extension

**DOI:** 10.3390/ijms23116009

**Published:** 2022-05-26

**Authors:** Kevin Adi Kurnia, Bonifasius Putera Sampurna, Gilbert Audira, Stevhen Juniardi, Ross D. Vasquez, Marri Jmelou M. Roldan, Che-Chia Tsao, Chung-Der Hsiao

**Affiliations:** 1Department of Bioscience Technology, Chung Yuan Christian University, Chung-Li 320314, Taiwan; kevinadik-adi@hotmail.com (K.A.K.); boni_bt123@hotmail.com (B.P.S.); gilbertaudira@yahoo.com (G.A.); stvn.jun@gmail.com (S.J.); 2Department of Chemistry, Chung Yuan Christian University, Chung-Li 320314, Taiwan; 3Department of Pharmacy, Faculty of Pharmacy, University of Santo Tomas, Manila 1015, Philippines; rdvasquez@ust.edu.ph; 4Research Center for the Natural and Applied Sciences, University of Santo Tomas, Manila 1015, Philippines; 5Faculty of Pharmacy, The Graduate School, University of Santo Tomas, Espana Blvd., Manila 1015, Philippines; mmroldan@ust.edu.ph; 6Department of Biological Sciences and Technology, National University of Tainan, Tainan 70005, Taiwan; 7Center of Nanotechnology, Chung Yuan Christian University, Chung-Li 320314, Taiwan; 8Research Center of Aquatic Toxicology and Pharmacology, Chung Yuan Christian University, Chung-Li 320314, Taiwan

**Keywords:** ImageJ, macro language, segmentation, *Tetrahymena*

## Abstract

Previous methods to measure protozoan numbers mostly rely on manual counting, which suffers from high variation and poor efficiency. Although advanced counting devices are available, the specialized and usually expensive machinery precludes their prevalent utilization in the regular laboratory routine. In this study, we established the ImageJ-based workflow to quantify ciliate numbers in a high-throughput manner. We conducted *Tetrahymena* number measurement using five different methods: particle analyzer method (PAM), find maxima method (FMM), trainable WEKA segmentation method (TWS), watershed segmentation method (WSM) and StarDist method (SDM), and compared their results with the data obtained from the manual counting. Among the five methods tested, all of them could yield decent results, but the deep-learning-based SDM displayed the best performance for *Tetrahymena* cell counting. The optimized methods reported in this paper provide scientists with a convenient tool to perform cell counting for *Tetrahymena* ecotoxicity assessment.

## 1. Introduction

Ciliated protozoa are unicellular eukaryotes commonly found in aquatic environments. They play an integral part in the community by connecting the food chain between bacteria and small phytoplankton to larger metazoa and zooplankton. Additionally, they are able to consume free organic material from the environment if necessary. These properties make the ciliated protozoan an appropriate organism to determine the health of an aquatic environment [1]. As a eukaryotic microorganism, *Tetrahymena* grows rapidly in the laboratory and divides every 2–3 h in the optimal condition, making it a superb experimental system for toxicological analysis [2,3,4,5]. By counting the cell number under different treatments and time courses, it is easy to monitor growth inhibition and assay the effect of a potential toxicant. The most common method to count *Tetrahymena* (or cultured cells in general) is using the hemocytometer, which is a popular laboratory technique for counting cells manually [6,7]. This method only requires basic equipment and a dozen microliters of cell culture but has a disadvantage of being low throughput, prone to human errors and subject to high variation. Meanwhile, advanced machineries, such as the Coulter counter, flow cytometer, image cytometer and microfluidic cytometer, have been applied to quantify cell numbers [8,9]. These expensive and specialized devices, however, may not be accessible for laboratories with limited resources or under an educational setting. Furthermore, they are not always suitable for day-to-day operations, since tweak and adjustment are often required or a larger volume of cells is needed.

Digital image analysis can facilitate manual counting to improve the efficiency and consistency. Software such as ImageJ provides an array of tools to assist the counting process, ranging from object counter, image editing and object calculation [10,11,12]. Available from the public domain, ImageJ can be adopted to manual counting or semi-automated counting using various built-in tools and community-supplemented plugins. For example, the Find Maxima tool of ImageJ determines the local intensity of the image, and users can adjust the prominence level for the object of interest (OOI) as a threshold for counting. Other preinstalled tools, Threshold and Analyze Particles, can be applied to highlight and count the objects according to the size and circularity threshold [10]. The Watershed tool, also preinstalled in ImageJ, applies the watershed algorithm to separate overlapping objects based on the edges. Furthermore, the setting and operating steps in ImageJ can be recorded and composed into a batch of commands, or macros, for easy execution in the subsequent, repeat analysis.

The built-in ImageJ-based tool by itself, however, encounters limitations for correctly segmenting overlapping objects. Advances in machine learning potentially provide a way to overcome this issue by transforming the segmentation problem into a pixel classification problem. Through labeling objects in the image and then making it a training set for the classifier, once the result has been obtained and improved by providing feedback, the classifier model can be used to classify similar images. Trainable WEKA segmentation comes preinstalled with the FIJI build of ImageJ and is convenient to use, since the training dataset is created by annotating the objects and background directly in ImageJ [13,14,15]. Another machine-learning method that can be integrated into the ImageJ platform is StarDist, which works by predicting the reference shape and is based on a neural network called U-Net [16]. A StarDist result is comparable to the state-of-the-art Mask R-CNN method but with advantages of being easier to train, use and requiring less tuning for the best result [17]. 

Since multiple tools for cell counting based on digital images are available, systemic assessment and documentation of their application should be useful for the research community. In this study, we investigated the utilization of three ImageJ-based methods [10], which are the find maxima method (FMM), particle analyzer method (PAM), and watershed method (WSM), to quantify the cell number of *Tetrahymena*. We also compared their results with those obtained from manual counting and from two machine-learning-assisted ImageJ processes: trainable WEKA segmentation (TWS) and StarDist (SDM). We established the workflow and composed the macro to simplify the operation.

## 2. Results

### 2.1. Overview of Our Setting and Analysis Pipeline for Tetrahymena Counting 

Cell counting can be used to calculate the cell concentration, which is often applied to indicate the growth rate of cells. The most common method to count cells is manual counting using a hemocytometer. A variety of software tools from the public domain can facilitate manual counting by analyzing digital images. Here, we sought to assess some of these available tools. We conducted cell counting of the ciliated protozoan *Tetrahymena*, established the workflow for five methods and compared their results. The experimental design is shown in Figure 1. We also supplied the Appendix A containing the steps we used for each tool in this study (Appendix A).

Our tested methods are divided into two groups: the ImageJ-based methods (FMM, PAM and WSM) and the machine-learning-based methods (TWS and SDM). All were mainly carried out on the ImageJ platform, since it is publicly accessible and the plugins for machine-learning-based methods have been developed. Analyses using the ImageJ-based methods started from image preprocessing for better detection and execution in both FMM and PAM. Afterward, the processed images were objected to the watershed algorithm to segment the overlapping cells (in WSM). These steps are bundled into one macro for ease of use (Appendix A). TWS and SDM were performed according to the developer’s guide.

### 2.2. Comparison of Tetrahymena Counting Performance between Known Methods

Through manual counting, the average cell of *Tetrahymena* from ten different images counting was 173.2 ± 8 cells/μL. Two ImageJ-based methods, FMM and PAM, yielded the exact cell count due to a shared image preprocessing. Using the manual counting as the reference, both FMM and PAM could recognize 156.4 ± 8.46 *Tetrahymena* with 90.30 ± 2.70% sensitivity (Table 1). In WSM, applying the watershed tool after image normalization could segment the cells better, thus reducing overlap incidents that were not counted in FMM and PAM. WSM recognized 172.4 ± 9.62 cells with 99.52 ± 2.33% sensitivity. Meanwhile, two machine-learning-based methods, TWS and SDM, could detect 157.5 ± 8.26 and 171.3 ± 8.57 cells, respectively (with 90.95 ± 2.88% and 98.89 ± 1.08% sensitivity, summarized in Table 1 and Figure 2A). Overall, all of the five methods could reasonably yield decent cell counts with the >90% sensitivity compared to the manual counting. Among the five methods tested, WSM and SDM showed superior results compared to the other three methods, with relatively low false negative, high count sensitivity, and more comparable to manual counting results (Table 1 and Figure 2A).

In addition to cell count, we also explored the potential application on measuring the average cell size and total area occupancy by cells from each image. Manual measurement showed an average cell size of 1027.76 ± 85.3 pixels and total area of 182,583.7 ± 18,556.0 pixels. The ImageJ-based PAM measured an average cell size of 1,145.0 ± 105.4 pixels and total area of 179,006.8 ± 18,148.7 pixels, and WSM measured an average cell size of 1,097.1 ± 105.5 pixels and total area of 187,745.3 ± 19,083.8 pixels. FMM was unable to calculate these endpoints because of its inherent limitation, i.e., that it is not for measuring the area of each object. Machine-learning-based methods showed an average cell size and total cell size of 1,194.4 ± 81.6 pixels and 187,965.4 ± 14,320.5 pixels for TWS and 987.9 ± 37.1 pixels and 169,264.4 ± 11,169.2 pixels for SDM, respectively (Table 1). Consistent with cell count data, we observed that SDM and WSM perform better than the other methods in measuring the average cell size (Figure 2B) and total area occupancy (Figure 2C), with more consistent values when compared to the manual measurement method.

To assess the consistency and performance of these five methods, we systematically investigated the correlation between the cell number, average cell size and total area obtained from every method compared to the manual counting. We used the Deming regression and calculated the correlation coefficient (r) for evaluation. Deming regression is a technique to fit a straight line to two-dimensional data where both variables are measured with error. Regression analysis of *Tetrahymena* cell count (summarized in Table 2) showed that all the methods tested in this study had a slope closely resembling the manual counting method (1.041–1.226, Figure 2D) and significant deviation from the zero-slope (*p* < 0.01 or smaller). In the machine-learning-based method, SDM showed a superior counting outcome compared to TWS due to a lower 95% confidence interval (CI) range, lower *p* value and higher correlation coefficient, which signifies that the value obtained from SDM has a lower variable range, which better fits the manual counting result compared to TWS. In the ImageJ-based method, WSM had a higher slope (1.226) compared to PAM and FMM (1.071). However, the counting values obtained from PAM and FMM were more variable and less consistent than WSM, as both the *p* value and correlation coefficient suggested, which means WSM is more suitable for *Tetrahymena* cell counting. Additionally, if we compare the best machine-learning-based and ImageJ-based methods, SDM as the machine-learning-based method comes on top for *Tetrahymena* cell counting.

The Deming regression of average cell size from SDM and WSM (Figure 2E) showed varied performance unlike the cell count result. In this endpoint, TWS showed the closest slope value to 1, which is 0.8372, followed by SDM at 0.3176. Meanwhile, ImageJ-based methods showed exceptionally high slope value at 3.637 and 5.141 for WSM and PAM, respectively. Even though TWS showed the closest slope value to 1, the *p* value (0.4846, ns) suggests that the TWS slope is not significantly different from zero, indicating that the data obtained by TWS are highly varied. WSM and PAM also showed a higher *p* value than TWS at 0.7257 (not significant) and 0.8129 (not significant), respectively. On the other hand, SDM showed significant difference with the zero slope through its *p* value (0.0378, *). Additionally, SDM also had the highest correlation coefficient at 0.6600, while the other methods we tested showed a relatively low correlation coefficient (WSM (0.1274), TWS, (0.2508) and PAM (0.08616)). However, a slope value of 0.3176 is far from 1, which means SDM is not suitable for measuring *Tetrahymena* cell size, although this method had superior performance in *Tetrahymena* cell counting. 

The Deming regression of total area yielded a similar result to the average cell size, showing the machine-learning-based method with low slope values far from 1 (0.5371 for SDM and 0.6634 for TWS, respectively). The ImageJ-based methods showed a better slope value (1.085 for WSM and 0.9470 for PAM). However, the statistical significance is not supported by the *p* values, which were 0.0055, ** for SDM; 0.3284, ns for WSM; 0.5833, ns for TWS; and 0.4048, ns for PAM. Similar to the average cell size result, the regression analysis of the total area endpoint suggests that none of the methods we tested were suitable for calculating the total area occupied by *Tetrahymena*. WSM and SDM showed the most comparable results to manual counting in terms of average cell size, while all of the methods showed similar results for total area occupancy to manual measurement. Deming regression was used as regression analysis to further confirm the methods we tested with manual counting/measurement. The regression analysis results showed WSM and SDM as the best methods for counting *Tetrahymena* cells; however, none of the methods showed comparable average cell size and total area occupancy to manual measurement. 

### 2.3. Effect of Various Tetrahymena Density on SDM and WSM Counting Performance

From the five methods tested, SDM proved itself as a well-performing machine-learning-based method, while WSM was the best ImageJ-based method for counting *Tetrahymena* cells. A further test was conducted to observe the counting performance of both methods in several different *Tetrahymena* cell densities. Afterward, the obtained cell count was pooled into one graph, which encompasses all the obtained data. Additionally, due to the unsatisfactory performance of the methods we tested in collecting/calculating the average cell size and total area occupancy, these endpoints were not pursued further.

With the addition of several cell densities spanning about an eight-fold difference around ~10^5^ cells/mL, we were still able to obtain satisfactory cell counting performance for both SDM and WSM methods. The slope values, which were 0.9630 for WSM and 1.002 for SDM, respectively, were close to the ideal slope value of 1, indicating both methods yielded a consistent and comparable result to manual counting. This conclusion was supported by their low *p* value and high correlation coefficient (Table 3 and Figure 3). The *p* value signifies there was a non-zero correlation between the methods we tested with manual counting; therefore, there was a linear association between the methods, as presented by the slope value that we obtained. Between the WSM and SDM results, SDM appeared to be a better method due to its closer slope value to 1 compared to the WSM value of 0.9630.

## 3. Discussion

Counting microorganisms is one of the main endpoints in environmental microbiology. It is mainly used to ascertain the effect of certain compounds of interest on microbes and microbial diversity [18,19]. While *Tetrahymena* is an important microbe in the aquatic environment [1], most methods for observing and counting *Tetrahymena* are manual, making the observation a tedious effort. The five methods we suggested here showed >90% sensitivity compared to manual counting. WSM and SDM showed more superior results compared to the other methods. The cell counting results are also supported by one-way ANOVA statistical test, which showed comparable results between WSM and SDM to manual counting. Average cell size and total area occupancy by *Tetrahymena* were also observed for additional endpoints. The result of these endpoints agreed with the cell count result, as they showed WSM and SDM to be the most compatible methods for measuring these parameters.

We also tested for Deming regression analysis, a common regression analysis used to compare methods with the same endpoints [20,21]. The Deming regression analysis supported the cell counting result, stating that all of the methods were compatible for counting *Tetrahymena* cells, with WSM and SDM as the best methods for cell counting. Meanwhile, none of the methods showed compatible results for average cell size and total area occupancy measurements, according to the Deming regression analysis result. Deming regression has a limitation where the equation is only applicable to values within the range; therefore, we added several cell densities to cover this limitation of Deming regression. Through the addition of samples with lower or higher cell densities (Figure A1), we found that the SDM and WSM methods can be applied for densities within a rather wide range, which is applicable for the routine experiment or cell assay.

Even though the overall result from the ImageJ-based method was not as good as the machine-learning-based method, ImageJ, as a major open-access platform, is still a powerful too, since other algorithms, such as the TWS and SDM, could be implemented to expand the versatile ImageJ function. The methods we tested in this study had their individual limitations and advantages when compared with each other. We found the PAM and TWS methods were unable to resolve overlapping objects in the object segmentation process. This could cause overlapping objects being counted as one object, reducing the cell counts and increasing the average size of each cell (Table 1 and Figure 2).

The built-in watershed tool in ImageJ (Figure 4A) was used to obtain better segmentation of overlapping cells. The watershed algorithm is known for separating different objects in an image. Therefore, it should be applicable for differentiating overlapped cells in our image. The WSM result showed promising cell count sensitivity of 99.5 ± 2.3% (Table 1). However, the Deming regression slope value and correlation coefficient showed WSM was less compatible for counting *Tetrahymena* cells than SDM. Meanwhile, SDM could separate overlapped *Tetrahymena* cells into several distinct cells compared to the other tested algorithms (Figure 4A). We also noticed that the WSM could not correctly recognize cells, which deviated from the regular shape, such as an unhealthy cell or a cell undergoing binary fission (Figure 4B). In conclusion, SDM performed better than WSM in image segmentation to obtain a better sensitivity in *Tetrahymena* cell counting.

All the ImageJ-based methods (FMM, PAM and WSM) have the advantage of being easier to learn than the other two methods. Creating a macro based on these methods can speed up the entire analysis process. Since these methods share a common workflow, they can even be compiled into one macro for convenience. There have been numerous publications that used TWS in ImageJ to process and segment images. For example, it has been used in the mineralogy field to study the distribution of crystal size and to develop a pap-smear analysis tool for detecting cancer cells through image detection [22]. TWS can process the images well in both studies, with some limitations in separating heterogeneities (foreign objects). Our study showed that it is possible for TWS to recognize smaller particles as heterogeneities and exclude the object in the final image. It was also able to recognize the difference of contrast between *Tetrahymena* cell images, which were not uniform. Although the machine-learning-based TWS could correctly detect *Tetrahymena* cells, WSM outperformed TWS in our study, since the former could achieve higher correlation coefficient, better *p* value (Table 2) and shorter workflow.

The final method we tested, SDM, showed the closest regression value to 1 (Figure 2 and Figure 3). The machine-learning-based SDM was designed to detect cell nuclei relying on star-convex polygons rather than the usual bounding box, as it better suits the natural roundish shape of cell nuclei [17,23]. Therefore, by design, SDM had a better performance in segmenting overlapping cells, which was demonstrated in our tests (Table 1 and Table 2, Figure 2, Figure 3 and Figure 4). However, the operational steps for SDM are more complicated compared to the other methods. First, training data set must be created by annotating similar images in QuPath, followed by training in Python, creating a model file, then applying the model to images on ImageJ platform. To simplify the entire process for *Tetrahymena* counting, we established the workflow of SDM and its model file (Appendix A), and this could serve as an entry point and exemplary for adopting SDM to analyze other planktons of a similar dimension and shape.

## 4. Material and Methods

### 4.1. Tetrahymena Cell Culture and Maintenance

The wild-type *Tetrahymena thermophila* CU428 strain was a gift from Dr. Meng-Chao Yao’s laboratory (Academia Sinica, Taipei, Taiwan). The culture was grown in a large beaker (PYREX^®^ 3000 mL) (Corning Inc., New York, NY, USA) containing SPP medium (1% proteose peptone, 0.1% yeast extract, 0.2% dextrose, 0.003% sequestrene) [24] and maintained at 26 °C. The cell was grown to the middle-log phase (1–2 × 10^5^ cells/mL) or late-log phase (6–7 × 10^5^ cells/mL) as the example for cell counting performance testing.

### 4.2. Tetrahymena Recording 

A volume of 100 μL of *Tetrahymena* cells was taken from the top region of the original culture using a micropipette and dispersed into a protozoan counting chamber with 10 × 10 grids (Zgenebio, Taipei, Taiwan) covered using a coverslip. Live *Tetrahymena* cells were counted from one grid of the protozoan counting chamber. As *Tetrahymena* are alive and move freely, it is assumed that they spread evenly across the protozoan counting chamber. Therefore, we alternated the recording at different time points within one grid and recorded several single grids within the counting chamber. The cell was observed using an upright microscope (ex20, SOPTOP, Taipei, Taiwan) equipped with a high-resolution 4K CCD (XP4K8MA, ToupTek, Zhejiang, China). Fields from one grid (corresponding to 1 μL volume) were recorded for 10 s at 4K resolution (3840 × 2160 pixels), 30 frames per second, with the 4× plan objective lens. The video was recorded and saved in the .mp4 format.

### 4.3. Image Processing 

The recorded video was converted to the .avi format from .mp4 file using VirtualDub2 software for use in ImageJ. Ten images were extracted from the 10 s video with the 1 s interval between images. The image stack was output for subsequent counting analysis and used as image for detection. In addition, 10 more images were randomly selected to compose the training dataset, which was used for machine-learning-based methods. The FIJI build of ImageJ (https://imagej.net/software/fiji/downloads, accessed on 21 May 2022) was used as the major image analysis platform and can be downloaded for free from the website. ImageJ-based methods firstly went through the preprocessing step, including binary conversion and size exclusion by thresholding using the Threshold tool and Analyze Particles tool. A binary mask with a threshold set at 300–Infinity was created to exclude foreign objects, which might cause false positives due to their contrast similarity to the object of interest. This process resulted in a binary image, which could increase the selection accuracy for PAM, FMM and WSM methods.

### 4.4. Find Maxima Method (FMM)

The find maxima method uses the Find Maxima tool provided by ImageJ. The prominence threshold was set to 50 for counting *Tetrahymena* cells. The result was saved in .tsv format through a macro command.

### 4.5. Particle Analyzer Method (PAM)

The particle analyzer method uses Analyze Particles tool provided by ImageJ. The size threshold was set as 300 to Infinity. Afterward, the Analyze Particles tool was used to save the data in the region-of-interest (ROI) manager, which also showed the result and summary of the cell counting. The data were saved in .tsv format through a macro command.

### 4.6. Watershed Segmentation Method (WSM)

The Watershed tool preinstalled on ImageJ was used to segment *Tetrahymena* in the image and then counted using the Analyze Particles method with the same-size threshold as PAM at 300–Infinity. The result was saved in .tsv format through a macro command.

### 4.7. Trainable WEKA Segmentation Method (TWS) 

Trainable WEKA segmentation comes preinstalled with the FIJI build of ImageJ. TWS implementation in FIJI was developed by Arganda-Carreras et al. [13]. The training dataset was created by annotating the object (*Tetrahymena* cells) and background (and non-cell particles) directly using the free-hand select tools of ImageJ. In this study, we used 30 images for training. From these images, we annotated 10 samples each for *Tetrahymena* and background per image. The training features we used in this study were the Gaussian blur, Sobel filter, membrane projections, difference of Gaussians, membrane projections, variance, median, mean, maximum, anisotropic diffusion, Laplacian and Kuwahara. After each training, manual checks for wrong classification and additional annotation were conducted. The training process was repeated until the desired results were achieved. Afterward, the model was saved as a classifier to detect ten images, which were prepared for detection.

### 4.8. StarDist Method (SDM) 

StarDist and its dependencies were installed on ImageJ using the provided installation guide in their ImageJ Wiki page (https://imagej.net/plugins/stardist, accessed on 25 August 2021). The training dataset was prepared using QuPath, as recommended by the StarDist developer (https://github.com/stardist/stardist, accessed on 25 August 2021). QuPath was used to annotate all *Tetrahymena* in the training image set, which contained 10 images, to obtain masks for training. Images used for the training dataset in StarDist method were the same images as those used in TWS. After annotating all *Tetrahymena*, jupyter notebook was used to run the commands for StarDist model training. The training process was performed using Anaconda with Python 3.7.0 implementation by following the provided tutorial in GitHub and running the export command to obtain a model (.zip) file, which was subsequently used in ImageJ to detect *Tetrahymena* cells from the 30 images prepared for detection. The training took around 12 h using GPU training on a computer with i7-9700k CPU, Nvidia GTX 1060 GPU and 32 GB of RAM. The probability/score threshold was set to 0.70 and the overlap threshold to 0.4 during ImageJ *Tetrahymena* detection. In addition, we also provided a supplementary tutorial video deposited on YouTube (https://www.youtube.com/watch?v=LPI8MWOvKn0&t=811s, accessed on 22 May 2022) for the detailed procedure for performing all the methods we used in this study.

### 4.9. Manual Counting 

Manual counting of *Tetrahymena* cells was performed using QuPath 0.2.3 (available online: https://qupath.github.io/, accessed on 10 September 2021). The area selection was performed using a brush tool. 

### 4.10. Sensitivity Calculation

The sensitivity of each method was calculated using the sensitivity equation:$$Sensitivity=\frac{True\;Positives}{True\;Positives+False\;Negatives}$$

The *True Positives* are cells that are detected by respective methods, while *False Negatives* are cells, which are not detected by the methods, or miscounted and overlapping cells, which should be counted as more than one.

### 4.11. Statistics and Reproducibility

Statistical analysis was performed using GraphPad Prism 8 (Graphpad Holdings, LCC, San Diego, CA, USA). One-way ANOVA was used to compare the result of the tested methods compared to manual counting at first; afterward, the Deming regression was used to further analyze the performance of each tested method compared to manual counting. 

## 5. Conclusions

The five methods we tested in this study showed acceptable performance for *Tetrahymena* cell counting, with their >90% sensitivity. However, some methods seemed to be less accurate compared to the others, as they were not able to distinguish or separate overlapping/clumped-up cells and non-cell objects in the images. At first, WSM and SDM showed the best sensitivity compared to PAM, FMM and TWS for cell counting. Through the Deming regression analysis, we found machine-learning-based SDM and ImageJ-based WSM to be the more suitable methods in their respective groups. In addition, we also tested two other endpoints, including the average cell size and total area occupancy. However, none of the methods showed ideal, comparable results to manual measurement. Therefore, we conclude that the currently established methods were the only suitable methods for *Tetrahymena* cell counting. More studies are still required to address the *Tetrahymena* cell size estimation issue in the future.

The regression analysis results showed that both SDM and WSM yielded accurate and consistent cell counts at cell density range from 10^5^ to ~10^6^ cells/mL (Figure 3 and Table 3). Machine-learning-implemented SDM showed the best compatibility and highest correlation with manual counting, with a slope able to reach 1.002. Thus, based on the five tested methods, we conclude that SDM, using StarDist for image segmentation and counting, is the best recommended method for *Tetrahymena* counting due to its accuracy and simplicity if a pretrained model is available, while WSM can be used due to its comparable result to SDM if there are no available pretrained models. In addition, the tools provided for counting *Tetrahymena* can be used freely due to ImageJ and QuPath being a free-to-use software. Our established workflow and supplemented model file also simplify the pretraining steps that allow scientists to run *Tetrahymena* cell counting program in a straightforward manner.

## Figures and Tables

**Figure 1 ijms-23-06009-f001:**
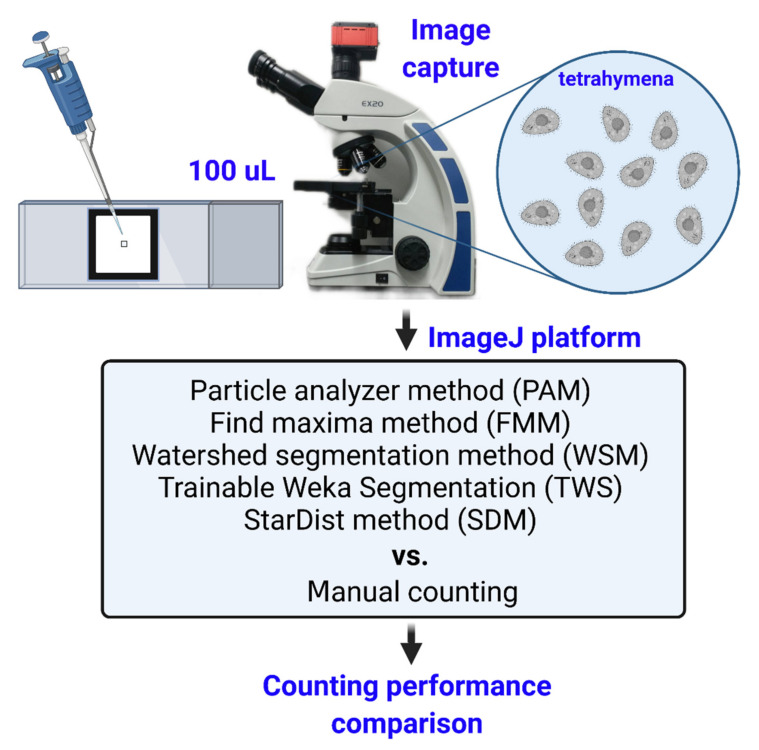
Experimental overview of analysis pipeline for *Tetrahymena* counting. First, 100 μL *Tetrahymena* samples were loaded into a protozoa counting chamber and covered with cover slide. Next, video recording was conducted for 10 s with 40× magnification. Later, videos were periodically output as 10 frames at 1 s interval. Finally, images were analyzed by five different cell counting methods and compared with manual counting method as the golden standard.

**Figure 2 ijms-23-06009-f002:**
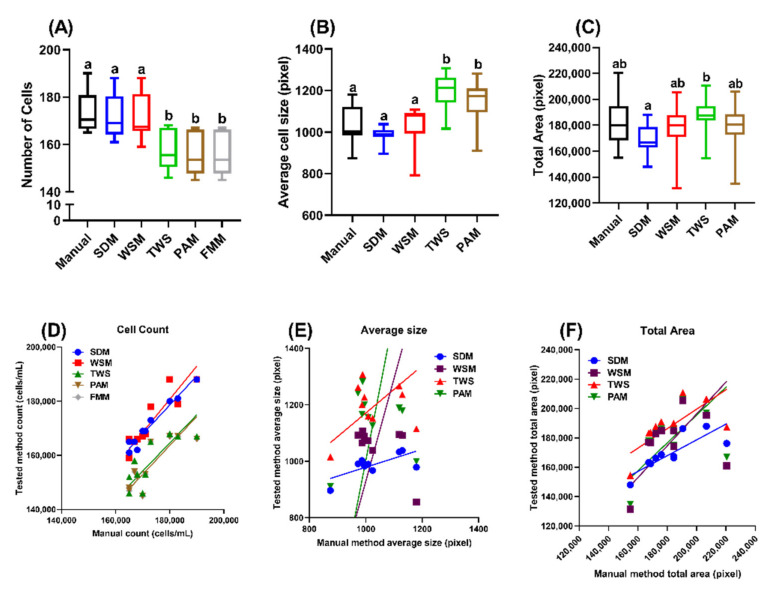
Assessment of the consistency and performance of five *Tetrahymena* counting methods. Comparison of the cell count (**A**), average cell size (**B**) and total area occupancy (**C**) result of *Tetrahymena* using five different methods for manual counting and measurement using one-way ANOVA. Different letter represents statistical difference (*p* < 0.05). Deming regression of the results obtained from five methods for manual counting and measurement of *Tetrahymena* cell count (**D**), average size (**E**) and total area (**F**). For each method, data obtained from ten image captures were used for comparison.

**Figure 3 ijms-23-06009-f003:**
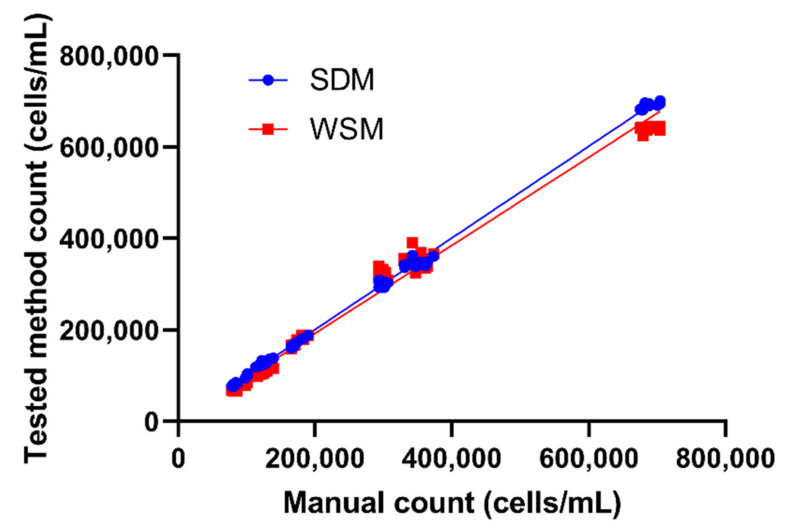
Deming regression of *Tetrahymena* cell count using either SDM or WSM on samples with a wide range of cell concentrations. For each method, cell counting data obtained from 80 different image captures were used for comparison.

**Figure 4 ijms-23-06009-f004:**
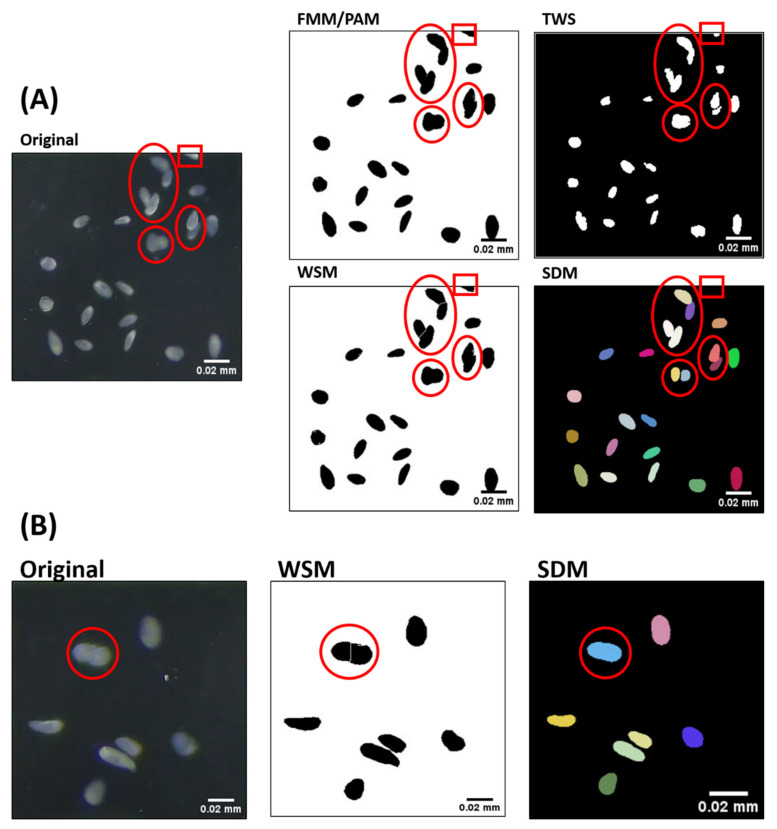
Images depicting segmentation results obtained from different *Tetrahymena* cell counting methods. (**A**) FMM/PAM and TWS showed the inability of segmenting overlapping *Tetrahymena* cells. In contrast, WSM showed limited segmentation, while SDM showed better segmentation. Segmented cells are marked with red circles, but misdetection on edge is marked with red rectangle. (**B**) Images showing segmentation error by WSM compared to SDM during cell division. *Tetrahymena* marked with red circle is undergoing binary fission, and it is still counted as single cell. This *Tetrahymena* was recognized as two cells by WSM due to the segmentation method.

**Table 1 ijms-23-06009-t001:** Comparison of cell counting performance for *Tetrahymena* between five different methods.

Method	Cell Count ± SD (Cells/μL)	False Negative	Count Sensitivity	Average Cell Size ± SD (Pixel)	Total Area ± SD (Pixel)
Manual	173.2 ± 8.0	-	-	1027.76 ± 85.3	182,583.7 ± 18,556.0
FMM	156.4 ± 8.0	16.8	90.3 ± 2.7%	Not available	Not available
PAM	156.4 ± 8.5	16.8	90.3 ± 2.7%	1145.0 ± 105.4	179,006.8 ± 18,148.7
WSM	172.4 ± 9.6	0.8	99.5 ± 2.3%	1097.1 ± 105.5	187,745.3 ± 19,083.8
TWS	157.5 ± 8.3	15.7	91.0 ± 2.9%	1194.4 ± 81.6	187,965.4 ± 14,320.5
SDM	171.3 ± 8.6	1.9	98.9 ± 1.1%	987.9 ± 37.1	169,264.4 ± 11,169.2

**Table 2 ijms-23-06009-t002:** Summary of Deming regression of *Tetrahymena* cell count using five different methods.

Group	SDM	WSM	TWS	PAM	FMM
Slope	1.073	1.226	1.041	1.071	1.071
95% Lower CL ^#^	0.8499	0.5038	0.1019	0.1850	0.1850
95% Upper CL	1.296	1.947	1.981	1.956	1.956
y-intercept	−14,537	−39,866	−22,865	−29,011	−29,011
95% Lower CL	−53,465	−162,036	−183,179	−179,390	−179,390
95% Upper CL	24,392	82,304	137,448	121,368	121,368
*p* value	<0.0001 (****)	0.0003 (***)	0.0054 (**)	0.0028 (**)	0.0028 (**)
Correlation coefficient (r)	0.9784	0.9096	0.8007	0.8320	0.8320

^#^ CL, Confidence limit. ** shows a *p* value of <0.01, *** shows a *p* value of <0.001, and **** shows a *p* value of <0.0001.

**Table 3 ijms-23-06009-t003:** Summary of Deming regression of *Tetrahymena* cell density using SDM and WSM at several different cell concentrations.

Group	SDM	WSM
Slope	1.002	0.963
95% Lower CL ^#^	0.9934	0.9346
95% Upper CL	1.01	0.9914
y-intercept	457.1	−300
95% Lower CL	−5825	−1135
95% Upper CL	5225	2049
*p* value	<0.0001 (****)	<0.0001 (****)
Correlation coefficient (r)	0.9934	0.9995

^#^ CL = Confidence limit. **** shows a *p* value of < 0.0001.

## Data Availability

Not applicable.

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
