# Peer review of "Performance Comparison of Five Methods for Tetrahymena Number Counting on the ImageJ Platform: Assessing the Built-in Tool and Machine-Learning-Based Extension"

_ijms, 2022, doi:10.3390/ijms23116009_

Round 1
Reviewer 1 Report
The manuscript by Kurnia et al. compares 5 methods for counting Tetrahymena versus the manual one. Only note I make. When comparing different methods, it is worth considering in addition to reliability and simplicity, also the economic aspect
Author Response
Comments and Suggestions for Authors
The manuscript by Kurnia et al. compares 5 methods for counting Tetrahymena versus the manual one. Only note I make. When comparing different methods, it is worth considering in addition to reliability and simplicity, also the economic aspect
We would like to thank the reviewers for their suggestion. We have adjusted the manuscript as suggested. First, we mentioned the software that we use are free to use as ImageJ and QuPath which are the main software for Tetrahymena cell counting is free-to-use. The plugins for ImageJ comes preinstalled/can be accessed freely from their page. Next, we also elaborated the operation steps of the 5 tested methods in both manuscript and procedure file. The StarDist Method (SDM) had the best reliability/accuracy compared to the other methods that we tested here, while its simplicity might be of concern due to the necessity of training, using a pretrained model is sufficient for similar images. If there is no pretrained model than we suggest the use of Watershed method (WSM) which has nearly comparable end result to SDM.
Reviewer 2 Report
Highly technical paper on some methods comparison in ImageJ. Not sure if it is relevant to the readers of this specific journal. I found it quite readable and really enjoy it. Statistics is very well done. Introduction sets the scene very well. Conclusions and Discussion section are well balanced and the take home message is clearly sent. For the specialists in this underestimated area and people working a lot with ImageJ is a good manuscript. I would suggest the acceptance of it.
Author Response
Comments and Suggestions for Authors
Highly technical paper on some methods comparison in ImageJ. Not sure if it is relevant to the readers of this specific journal. I found it quite readable and really enjoy it. Statistics is very well done. Introduction sets the scene very well. Conclusions and Discussion section are well balanced and the take home message is clearly sent. For the specialists in this underestimated area and people working a lot with ImageJ is a good manuscript. I would suggest the acceptance of it.
The authors would like to thank the reviewers for their response. The authors understood reviewers point of view regarding the relevancy of this manuscript in this specific journal, however, the authors believed the comparison of these five methods will provide a deeper understanding in cell segmentation analysis especially for Tetrahymena and in the future. This study might inspire the development of similar methods which are capable of differentiating cells with different visual properties due to differences in molecular properties.